# *N*-Glycan Modifications with Negative Charge in a Natural Polymer Mucin from Bovine Submaxillary Glands, and Their Structural Role

**DOI:** 10.3390/polym13010103

**Published:** 2020-12-29

**Authors:** Jihye Kim, Byoungju Lee, Junmyoung Lee, Minkyoo Ji, Chi Soo Park, Jaeryong Lee, Minju Kang, Jeongeun Kim, Mijung Jin, Ha Hyung Kim

**Affiliations:** 1Biotherapeutics and Glycomics Laboratory, College of Pharmacy, Chung-Ang University, 84 Heukseok-ro, Dongjak-gu, Seoul 06974, Korea; non1172@naver.com (J.K.); bjlee01@gmail.com (B.L.); 123jun123@naver.com (J.L.); changlyong2@naver.com (M.J.); ack11111@naver.com (C.S.P.); ioi_1007@naver.com (J.L.); alswn3746@naver.com (M.K.); kje12191219@naver.com (J.K.); zzini0620@naver.com (M.J.); 2Department of Global Innovative Drugs, Graduate School of Chung-Ang University, 84 Heukseok-ro, Dongjak-gu, Seoul 06974, Korea

**Keywords:** bovine submaxillary mucin, *N*-glycan, sulfate modification, structural role, functionality

## Abstract

Bovine submaxillary mucin (BSM) is a natural polymer used in biomaterial applications for its viscoelasticity, lubricity, biocompatibility, and biodegradability. *N*-glycans are important for mucin stability and function, but their structures have not been fully characterized, unlike that of *O*-glycans. In this study, BSM *N*-glycans were investigated using liquid chromatography-tandem mass spectrometry. The microheterogeneous structures of 32 *N*-glycans were identified, and the quantities (%) of each *N*-glycan relative to total *N*-glycans (100%) were obtained. The terminal *N*-acetylgalactosamines in 12 *N*-glycans (sum of relative quantities; 27.9%) were modified with mono- (10 glycans) and disulfations (2 glycans). Total concentration of all sulfated *N*-glycans was 6.1 pmol in BSM (20 µg), corresponding to 25.3% of all negatively charged glycans (sum of present *N*-glycans and reported *O*-glycans). No *N*-glycans with sialylated or phosphorylated forms were identified, and sulfate modification ions were the only negative charges in BSM *N*-glycans. Mucin structures, including sulfated *N*-glycans located in the hydrophobic terminal regions, were indicated. This is the first study to identify the structures and quantities of 12 sulfated *N*-glycans in natural mucins. These sulfations play important structural roles in hydration, viscoelasticity control, protection from bacterial sialidases, and polymer stabilization to support the functionality of BSM via electrostatic interactions.

## 1. Introduction

Asparagine-linked (*N*-) glycans containing negatively charged moieties, such as sialic acid, phosphate, or sulfate groups, frequently have crucial roles in the function of glycoproteins [1]. The presence of sialylated *N*-glycans is related to the stability and half-life of glycoproteins [2]. Phosphorylated *N*-glycans, including the mannose-6-phosphate group, are responsible for the efficacy of enzyme replacement therapy [3], and sulfated *N*-glycans are essential for electrostatic interactions and signal transduction pathways [4]. However, these *N*-glycans are difficult to analyze because of the labile nature of their charged moiety, their heterogeneous composition and structure, and their small proportions relative to total glycan quantities [5,6].

*N*-glycans are released by the *N*-glycosidase digestion of glycoproteins. The reducing ends of the released *N*-glycans are labeled with fluorophores, such as 2-aminobenzamide (AB), 2-aminobenzoic acid, 2-aminopyridine, and procainamide (ProA) [7], as *N*-glycans do not contain chromophores or fluorophores. Fluorophore-labeled *N*-glycans have been analyzed using ultraperformance liquid chromatography (UPLC), capillary electrophoresis, matrix-assisted laser desorption/ionization (MALDI) time-of-flight (TOF) mass spectrometry (MS), and liquid chromatography (LC)-tandem MS (MS/MS) [8]. AB is the most frequently used fluorophore for the labeling of glycans [8,9]; however, ProA-labeling improves the ability to identify glycans using electrospray ionization (ESI)-MS [10], enhances MS ionization efficiency approximately 10–50 times [9], and minimizes the ionization deviation of charged glycans by increasing hydrophobicity [11].

Mucus is a hydrogel comprised of electrolytes, lipids, growth factors, enzymes, mucins, and water (~95%) [12]. Mucins are highly Ser/Thr-linked (*O-*) glycosylated macromolecular polymers [13] consisting of hydrophobic terminal regions and hydrophilic *O*-glycosylated central domains [14]. Hydrated mucins are formed in polymer networks by self-association [15]. Mucin research has mainly focused on the effects of *O*-glycans on mucin structure and function [13], including the generation of extended conformations, binding of pathogens to reduce their growth, prevention of protease cleavage, and polymerization of mucins [12]. Mucin *O*-glycans consist of fucose (Fuc), galactose (Gal), *N*-acetylglucosamine (GlcNAc), *N*-acetylgalactosamine (GalNAc), and sialic acid [16]. Mucin *N*-glycosylation has been studied less than the abundant *O*-glycosylation. However, *N*-glycans are important in mucin biosynthesis, folding, stability, function, and cell surface localization [17,18].

Natural polymer mucin from bovine submaxillary glands (bovine submaxillary mucin; BSM) is a macromolecular (4 MDa; as determined by light-scattering) glycoprotein with unique properties, such as viscoelasticity, lubricity, biocompatibility, biodegradability, and non-toxicity [19,20]. The viscoelastic and lubricating properties of gel-forming mucins (including BSM) depend on mucin network formation conditions such as concentration, pH, temperature, ionic strength, metal ions, disulfide bonds, hydrogen bonding, charge, glycan composition, and hydration degree [12,15,21]. Specifically, electrostatic interactions between charged mucin moieties and charged surfaces have been reported that could strongly enhance or reduce the physical adsorption of mucins [14].

BSM is used in various biomaterial and biomedical applications [22,23,24] including the adsorption of the non-glycosylated regions of other biomolecules, protection against bacterial and fungal infections [25], coating of pathogens to reduce their growth [26], hydrophobic drug solubilization, artificial tear solution components, and sustained release drug delivery systems [12,19]. Recently, BSM glycosylation has been characterized to identify the glycan structures [19,27,28] and glycosylation sites [29].

In this study, the structures and quantities of *N*-glycans of BSM labeled with ProA were identified using UPLC and LC-ESI-high-energy collisional dissociation (HCD)-MS/MS to confirm the presence of negatively charged *N*-glycans, even those in small quantities. The mucin structure, including the presence of *N*-glycan structures and the structural roles of the identified *N*-glycans for BSM functionality, was ascertained.

## 2. Materials and Methods

### 2.1. N-Glycan Preparation

BSM (Type I-S) was purchased from Sigma-Aldrich (St. Louis, MO, USA) and treated with *N*-glycosidase (PNGase F; New England Biolabs, MA, USA) according to the manufacturer’s instructions. The released *N*-glycans were separated from the remaining peptides and salts with a graphitized carbon cartridge [30] and derivatized with ProA (Sigma-Aldrich) [31]. Briefly, a ProA-derivatizing reagent made with sodium cyanoborohydride (9.7 mg) and ProA (16.2 mg) in 150 µl of 30% acetic acid in dimethyl sulfoxide (*v/v*) was added to the released *N*-glycans. Excess fluorophores were removed using microcrystalline cellulose (Sigma-Aldrich, St. Louis, MO, USA) column chromatography.

### 2.2. N-Glycan Analysis Using UPLC

ProA-labeled *N*-glycans were analyzed using the Waters ACQUITY UPLC H-class system equipped with a Waters ACQUITY UPLC fluorescence detector. Hydrophilic interaction LC-based separation was performed using an ACQUITY UPLC Glycan BEH amide column (1.7 μm, 2.1 × 150 mm; Waters, Milford, MA, USA). Solvents A and B were 50 mM ammonium formate (pH 4.4) and 100% acetonitrile, respectively. The gradient conditions allowed the linear changes of solvent B from 75% to 50% at a flow rate of 0.5 mL/min for 46.5 min. The excitation and emission wavelengths were set at 310 nm and 370 nm, respectively.

### 2.3. N-Glycan Analysis Using LC-ESI-HCD-MS/MS

LC-ESI-HCD-MS/MS was performed using an Ultimate 3000 LC system (Thermo Scientific, Waltham, MA, USA) coupled with a mass spectrometer (Q Exactive Hybrid Quadrupole-Orbitrap; Thermo Scientific). The same column and gradient were used for LC-ESI-HCD-MS/MS as for UPLC. The MS measurement was performed in positive ion mode over a range of *m/z* 500, and the HCD-MS/MS spectra were acquired over *m/z* 100 to observe *N*-glycan-generated fragment ions. The detailed mass spectrometer settings for MS analyses are discussed below. The full scan (*m/z* 500–3000) used 70,000× resolution with an automatic gain control target of 1 × 10^6^ ions and a maximum injection time of 100 ms. Data-dependent MS/MS was performed in a “Top 5” data-dependent mode using the following parameters: 17,500× resolution, 2 × 10^5^ automatic gain control target, and 50 ms maximum injection time. The source ionization was conducted with 3.5 kV spray voltage, 320 °C capillary temperature, 300 °C probe heater, and an S-Lens level of 50. The data acquisition was performed with an Xcalibur 2.1 (Thermo Fisher Scientific). Mass error was calculated as follows: ((observed mass − calculated mass)/(calculated mass)) × 10^6^ [3]. All analytical experiments were repeated three times with identical LC chromatograms and MS ion chromatograms.

### 2.4. Relative Quantity of N-Glycans

The quantity (%) of each *N*-glycan relative to total *N*-glycans (as 100%) was determined from the relative extracted-ion chromatogram (EIC) areas for all observed charge states of each *N*-glycan relative to the total EIC area for all *N*-glycans generated by LC-ESI-HCD-MS/MS.

### 2.5. Absolute Concentration of N-Glycans

The absolute concentration (mol) of each *N*-glycan in BSM (20 µg) was determined from the fluorescence intensity of two distinct *N*-glycan peaks (negatively charged and neutral glycans) in the UPLC chromatogram using a linear calibration curve generated from the ProA concentrations (0.1–12.5 pmol) with two-fold serial dilutions. The other concentrations were calculated using the relative quantities from the concentrations of the distinct peaks.

## 3. Results

### 3.1. UPLC and LC-ESI-HCD-MS/MS Analysis of N-Glycans

BSM protein purity was confirmed using SDS-PAGE (data not shown), as described in our previous reports [19,29]. The *N*-glycans released from BSM were derivatized with ProA and analyzed using UPLC and LC-ESI-HCD-MS/MS. All of the analytical experiments were repeated three times and they yielded identical chromatograms and MS data.

*N*-glycan peaks were obtained from UPLC chromatograms (Figure 1A) and the total ion chromatogram (TIC; Figure 1D) corresponding to 32 *N*-glycan structures (Appendix A), which were identified by the EICs of two diagnostic oxonium ions: [Sulfate(-SO_4_;S)_1_*N*-acetylhexosamine(HexNAc)_1_ + H]^+^ (calculated *m/z* = 284.0435; Figure 1B) and [ProA-Fuc_1_HexNAc_1_ + H]^+^ (calculated *m/z* = 587.3287; Figure 1C). 

A total of 32 peaks from *N*-glycans were identified in the UPLC chromatogram (Figure 1A), including 12 peaks from *N*-glycans with sulfation (Figure 1B) and 18 peaks from *N*-glycans with fucosylation (Figure 1C). These peaks were numbered in the EICs with corresponding numbers in Figure 1A. Seven peaks (peaks 5, 8, 13, 16, 20, 24, and 27) could be observed both in Figure 1B,C. 

### 3.2. Structural Analysis of N-Glycans

*N*-glycan structures were identified using LC-ESI-HCD-MS/MS in the following order: (1) a full-scan mass spectrum of the precursor ion with its isotopes; (2) MS/MS spectrum for the fragment ions generated from the precursor ion, as HCD fragmentation cleaves the glycosidic bonds of glycan; and (3) a full-scan mass spectrum for the multiple-charged precursor ion.

The fragmentation spectra for the *N*-glycan structural analysis are presented in Figure 2 and Appendix A. The common oxonium ions for the *N*-glycans, [HexNAc + H]^+^ (observed *m/z* = 204.0867 and 204.0868; mass errors = 0.0 and 0.5 ppm) and [ProA-HexNAc + H]^+^ (observed *m/z* = 441.2714, 441.2710, and 441.2703; mass errors = 1.6, 0.7, and −0.9 ppm), were observed in the HCD-MS/MS spectra (Figure 2).

Figure 2A shows the MS/MS spectrum of the sulfated *N*-glycan, which was determined by identifying a precursor ion (observed *m/z* = 991.3793; mass error = −2.6 ppm; peak 18 in Table 1) and the characteristic oxonium ion-including sulfate [S_1_HexNAc_1_ + H]^+^ (observed *m/z* = 284.0430; mass error = −1.8 ppm). We observed fragment ion peaks that monosaccharides consecutively separated from [ProA-HexNAc_4_Hexose(Hex)_4_ + H]^+^ (observed *m/z* = 1698.7222), such as [ProA-HexNAc_4_Hex_3_ + H]^+^ (observed *m/z* = 1536.6671) and [ProA-HexNAc_3_Hex_3_ + H]^+^ (observed *m/z* = 1333.5874).

Figure 2B shows the structure of oxonium ions, including sulfate [S_1_HexNAc_1_ + H]^+^ (observed *m/z* = 284.0441; mass error = −2.1 ppm) and core-Fuc [ProA-Fuc_1_HexNAc_1_ + H]^+^ (observed *m/z* = 587.3280; mass error = −1.2 ppm). Core-Fuc in *N*-glycan generally migrated to the end of the glycan during the analyses [31]. However, the MS/MS spectra indicated that the fragment ion [ProA-Fuc_1_HexNAc_1_ + H]^+^ was suggestive of a core-fucosylated *N*-glycan with no migration. Additionally, the precursor ion [ProA-S_1_Fuc_1_HexNAc_4_Hex_4_ + H]^+^ (observed *m/z* = 962.8702; mass error = −0.9 ppm; peak 16 in Table 1) was identified by confirming the fragment ion of (1) [ProA-Fuc_1_HexNAc_3_Hex_4_ + H]^+^ (observed *m/z* = 1641.7054), (2) [ProA-Fuc_1_HexNAc_2_Hex_4_ + H]^+^ (observed *m/z* = 1479.6472), and (3) [ProA-Fuc_1_HexNAc_3_Hex_3_ + H]^+^ (observed *m/z* = 1438.6188). Thus, this MS/MS spectrum was determined as *N*-glycan containing both sulfate and core-Fuc.

The MS/MS spectrum of *N*-glycan containing core-Fuc indicated the presence of oxonium ions, including core-Fuc [ProA-Fuc_1_HexNAc_1_ + H]^+^ (observed *m/z* = 587.3292; mass error = 0.9 ppm), precursor ions (observed *m/z* = 1003.9183; mass error = −0.8 ppm; peak 31 in Table 1), and their fragment ion peaks (Figure 2C). The other MS/MS spectra of the identified *N*-glycans are indicated in the Appendix A.

### 3.3. Structure and Quantification of N-Glycans

Table 1 summarizes the structure and relative quantity of 32 BSM *N*-glycans confirmed by the fragmentation patterns in the MS/MS spectra, which revealed that oxonium ions depend on their constituents. The *N*-glycans were detected as 6 singly and 26 doubly charged ion forms, all observed with a high accuracy (<5.0 ppm).

The area of each peak of the UPLC chromatogram usually indicates the glycan quantity. However, in this study, the quantity (%) of each *N*-glycan relative to the total *N*-glycan amount (as 100%) was characterized according to the EIC rather than the UPLC chromatogram, as there were some overlapping peaks in the UPLC chromatogram, unlike in the EIC extracted as a single peak.

The 32 *N*-glycans included 12 sulfated (27.9%, sum of relative quantities of each glycan) and 20 non-sulfated or non-core-fucosylated (72.1%) *N*-glycans. Their characterization depended on sulfation and/or core-fucosylation, such as 5 sulfated *N*-glycans (S_1~2_Fuc_0_HexNAc_4~6_Hex_3~5_; 6.2%), 7 sulfated and core-fucosylated *N*-glycans (S_1~2_Fuc_1~2_HexNAc_4~6_Hex_3~4_; 21.7%), 11 core-fucosylated *N*-glycans (S_0_Fuc_1~2_HexNAc_2~6_Hex_3~5_; 50.9%), and 9 non-sulfated and non-fucosylated *N*-glycans (S_0_Fuc_0_HexNAc_2~6_Hex_3~6_; 21.2%).

The 12 sulfated *N*-glycans consisted of 10 mono- and 2 di-sulfated *N*-glycans at the terminal GalNAc, and the 7 sulfated *N*-glycans were core-fucosylated at the innermost GlcNAc. No sialylated or phosphorylated *N*-glycans were identified.

The MS ionization efficiency of *N*-glycans was improved by ProA-labeling in this study, and 65 molecular ion peaks were obtained (Figure 3). However, 32 *N*-glycans (relative quantity >0.7%) were confirmed by the MS/MS spectra. No fragmentation pattern was obtained for the other 33 molecular ion peaks (33–65 in Figure 3) in each MS/MS spectrum; therefore, these results were not included in the present study.

## 4. Discussion

BSM is a polymer with sialylated *O*-glycosylation, which has an important role in the formation of polymer networks [15,32], and sulfated *O*-glycans have been previously reported to provide protection against foreign bacterial sialidases [12]. Moreover, *N*-glycans of mucin has been reported to be important for stability, function, and polymerization [17,18]. However, the structures of mucin *N*-glycans have not been fully characterized, unlike that of *O*-glycans.

In this study, the structures of 32 *N*-glycans (including 12 sulfated and 18 core-fucosylated *N*-glycans) and the quantities (%) of each *N*-glycan relative to total *N*-glycans (100%) of BSM were identified by LC-ESI-HCD-MS/MS. *N*-glycans were labeled with ProA to enhance mass ionization efficiency, minimize ionization deviation of charged glycans, and inhibit core-fucose migration to the terminal site [10,31]. Of the 32 *N*-glycans, 15 unreported *N*-glycans were not identified without ProA-labeling approach. These included 5 sulfated (6.2%), 5 sulfated and core-fucosylated (9.6%), 3 core-fucosylated (4.6%), and 2 nonsulfated and non-fucosylated (3.5%) *N*-glycans. The relative quantity of each unreported *N*-glycan ranged from 0.7 to 2.6% (average 1.6% each), and the sum of their relative quantities was 23.9% of the total *N*-glycans.

Sulfation reportedly modifies the terminal GalNAc of *N*-glycans [33]. The importance of sulfation in mucin has been demonstrated in previous studies focusing on water-holding capacity [32] and the protection of degradation from bacterial sialidases [12]. Additionally, glycan length and the degree of sulfation in glycans might control mucin viscoelasticity [34]. Moreover, the removal of mucin sulfate groups by treatment with sulfatase type H-1 from *Helix pomatia* reduced its adsorption and lubricity, suggesting that the functionality of mucins is closely related to their glycosylation pattern [14].

Core-fucosylated *N*-glycans in glycoproteins are important in the regulation of protein functions involving receptor-binding affinity, malignant tumors, immunity, and disease biomarkers [35]. Core-fucosylated *N*-glycans are also reportedly crucial for the efficacy and safety of biotherapeutic glycoproteins [31]. However, this core-fucosylation is difficult to analyze because of the migration of the core-Fuc at the innermost GlcNAc in the *N*-glycans to the terminal site when using MS/MS, but ProA reportedly inhibits this migration [31].

Schematic illustrations of the mucin structures based on the present results and previous reports [12,14] are shown in Figure 4. The *N*-glycans, including the negatively charged sulfated *N*-glycans identified in this study, are located in the hydrophobic terminal regions (Figure 4A) [12,19].

The negatively charged glycans or positively charged amino acids (containing Lys, Arg, and His) in mucin might enhance its adsorption onto positive or negative surfaces in an aqueous environment, respectively. Sulfated *N*-glycans might contribute to the adsorption (or desorption) onto positive (or negative) surfaces due to attractive (or repulsive) forces, as well as hydration with water-holding and protection of degradation from foreign bacterial sialidases in mucin [12].

The C-terminal Cys-rich part (including the Cys knot) of mucin forms disulfide-bonded mucin dimers and multimers [36]. The present results suggest that the physicochemical characterization (e.g., pH sensitivity and stability) of BSM is derived from the sulfate (pKa = 1) groups in *N*-glycans as well as the reported sialic acid (pKa = 2.6) with some sulfate groups in *O*-glycans and charged amino acids [37]. Finally, sulfated *N*-glycans may structurally affect mucin mobility due to inter- and intramolecular electrostatic interactions, and associative polymer networks through self-association could form and stabilize together with hydrogen bonding (Figure 4B).

The structural modification of glycans has been identified and characterized using radioactive-isotope labeling, nuclear magnetic resonance spectroscopy, and LC-MS/MS [27,38,39]. A comprehensive assessment of the use of AB, ProA, aminoxyTMT (Thermo Scientific), RapiFluor-MS (Waters), and permethylation for *N*-glycan analysis was recently reported [7]. RapiFluor-MS labeling was suggested to provide the highest MS signal enhancement for neutral glycans; however, these increased ratios were not significantly different (<10 times). The aminoxyTMT and RapiFluor-MS kits involve a short sample preparation time but are more expensive than the others. The analysis of sulfated *N*-glycans using permethylation to effectively remove negative charges has also elucidated increases in hydrophobicity and ionization. However, permethylation generates heterogeneous *N*-glycans as a result of non-specific processing, and large methylated *N*-glycans are difficult to elute from a column [7].

The stoichiometric ratio of ProA to glycan is reportedly 1:1 [38]. Therefore, glycan concentrations were estimated based on the fluorescence intensities of the negatively charged (peak 13 in Figure 1) and neutral (peak 2 in Figure 1) glycan peaks using a linear calibration curve (r^2^ = 0.99) generated from the ProA concentrations (0.1–12.5 pmol) (data not shown). The concentration range of each unreported *N*-glycans was 0.2–0.6 pmol. The total concentration, which was the sum of 12 negatively charged (6.1 pmol) and 20 neutral (15.4pmol) *N*-glycans, was 21.5 pmol in 20 µg of BSM.

Recently, *O*-glycans of BSM have been quantitatively reported to be in total concentrations of 1.1 pmol per 1 μg of BSM, and the negatively charged sialylated *O*-glycans occupy 81.9% of total *O*-glycans [29]. From the present and reported results, it is reasonable to suggest that sulfated *N*-glycans account for 14.0% of total glycans (sum of the present total *N*-glycans and reported total *O*-glycans) [6.1 pmol/(21.5 pmol + 22.0 pmol)], and these were 25.3% of negatively charged *N*-glycans per total negatively charged glycans (sum of the present negatively charged *N*-glycans and reported negatively charged *O*-glycans) [6.1 pmol/(6.1 pmol + 18.0 pmol)].

There have been some reports of *N*-glycans in human and rat mucins. According to MALDI-TOF MS analysis, human corneal mucin MUC16 has *N*-glycans containing core-Fuc and sialic acid [17], whereas rat sublingual mucin contains only hybrid *N*-glycans [40]. The human mucin MUC1 also contained neutral *N*-glycans based on MALDI-TOF MS, gas chromatography-MS, and collisionally activated dissociation-ESI-MS/MS [41]. BSM was recently reported as the most appropriate complement for an extremely complex biocolloid human salivary mucin for use in biomaterial coatings [42] and is an optimal commercially available mucin source because of its similarity to human saliva in terms of surface adsorption and lubrication [33]. However, none of these reports provide information on the presence of sulfation in *N*-glycans, and further studies are needed to identify sulfated *N*-glycans.

## 5. Conclusions

The present study identified 12 sulfated *N*-glycans with negatively charged moieties in BSM. The sulfate modifications were observed in 27.9% (6.1 pmol total concentration) of total *N*-glycans in BSM (20 µg), and these occupy 14.0% of total glycans (sum of the present *N*-glycans and reported *O*-glycans [29]) and 25.3% of all negatively charged *N*- and reported *O*-glycans [29]. Together with the reported sialylation of *O*-glycans, the negatively charged sulfations in *N*-glycans play important structural roles in hydration [32], viscoelasticity control [32], and polymer network stabilization for maintaining gel-forming mucin functionality because of their electrostatic interactions. These *N*-glycans were located in the hydrophobic terminal regions of mucin, and we indicated the mucin structures, which will be useful for understanding the functionality of mucins and their application as biomaterials. This is the first study to structurally characterize and quantify 12 microheterogeneous sulfate modifications in *N*-glycans of natural mucins.

## Figures and Tables

**Figure 1 polymers-13-00103-f001:**
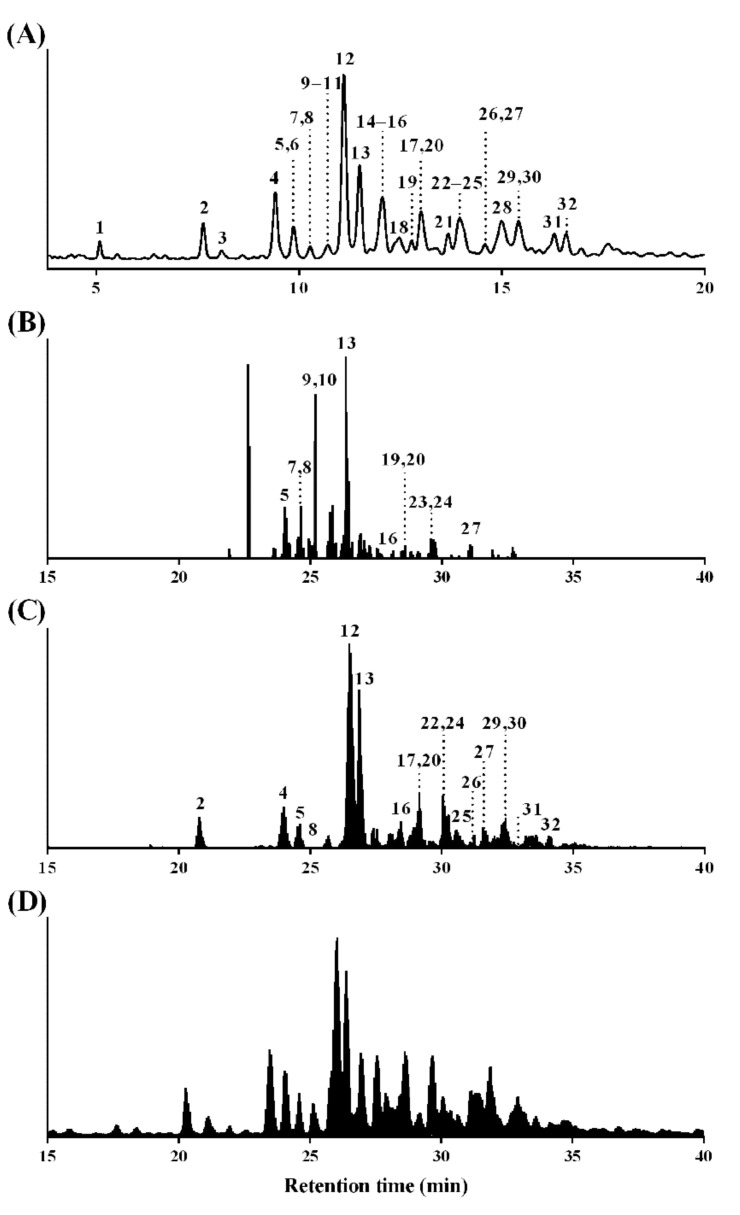
(**A**) Ultraperformance liquid chromatography (UPLC) chromatogram, (**B**) extracted-ion chromatogram (EIC) of diagnostic oxonium ion for sulfation ([S_1_HexNAc_1_ + H]^+^, *m/z* 284.0435), (**C**) EIC of diagnostic oxonium ion for core-fucosylation ([ProA-Fuc_1_HexNAc_1_ + H]^+^, *m/z* 587.3287), and (**D**) total ion chromatogram (TIC) of ProA-labeled *N*-glycans from bovine submaxillary mucin (BSM). The peak numbers correspond to *N*-glycans identified using LC-ESI-HCDMS/MS.

**Figure 2 polymers-13-00103-f002:**
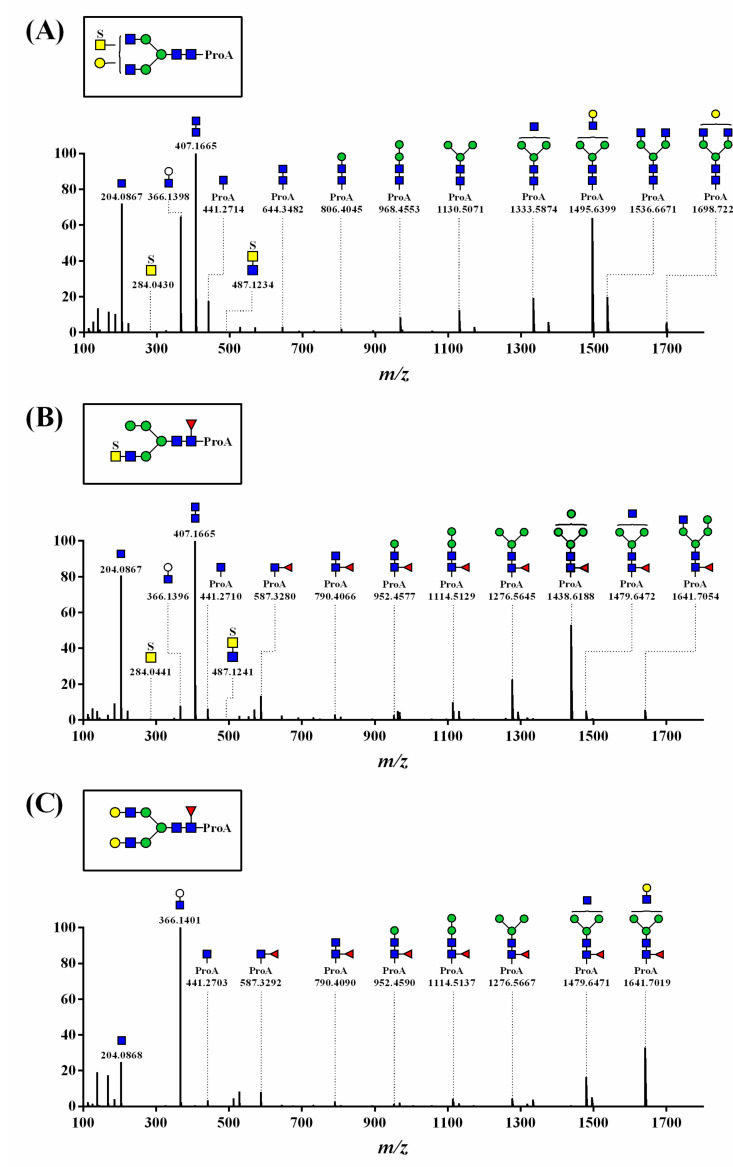
HCD-MS/MS spectra of (**A**) sulfated (peak 18), (**B**) sulfated and core-fucosylated (peak 16), and (**C**) core-fucosylated (peak 31) *N*-glycans with ProA-labeling detected as doubly charged precursor ion [M+2H]^2+^. Peak numbers correspond to the respective peak numbers in Table 1 and Figure 1. Symbols: S, sulfate; ▲, Fuc; ■, GlcNAc; ■, GalNAc; ●, Gal; ●, Man.

**Figure 3 polymers-13-00103-f003:**
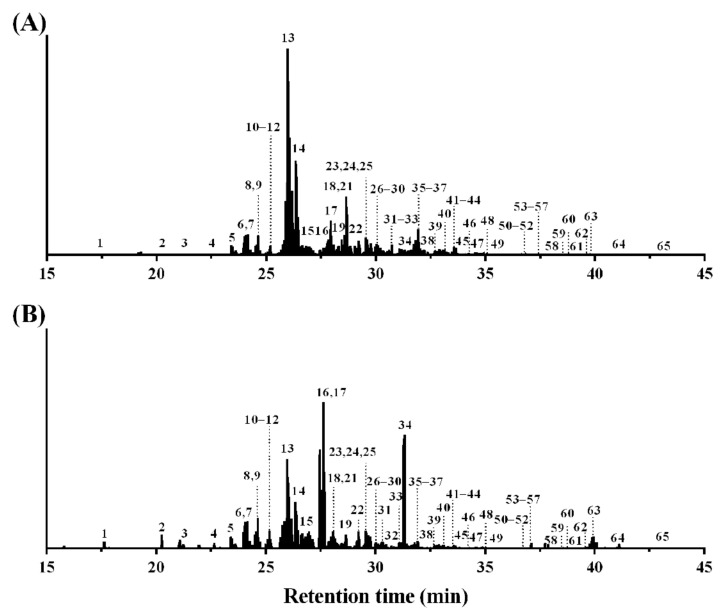
EIC of diagnostic oxonium ion for (**A**) ([HexNAc_1_ + H]^+^, *m/z* 204.0867) and (**B**) ([ProA-HexNAc_1_ + H]^+^, *m/z* 441.2708) of ProA-labeled *N*-glycans from BSM. The number of precursor ion peaks (1–32; relative quantity > 0.7%) and molecular ion peaks (33–65; not included in the present results) were obtained from full scan mass spectra using LC-ESI-HCD-MS/MS.

**Figure 4 polymers-13-00103-f004:**
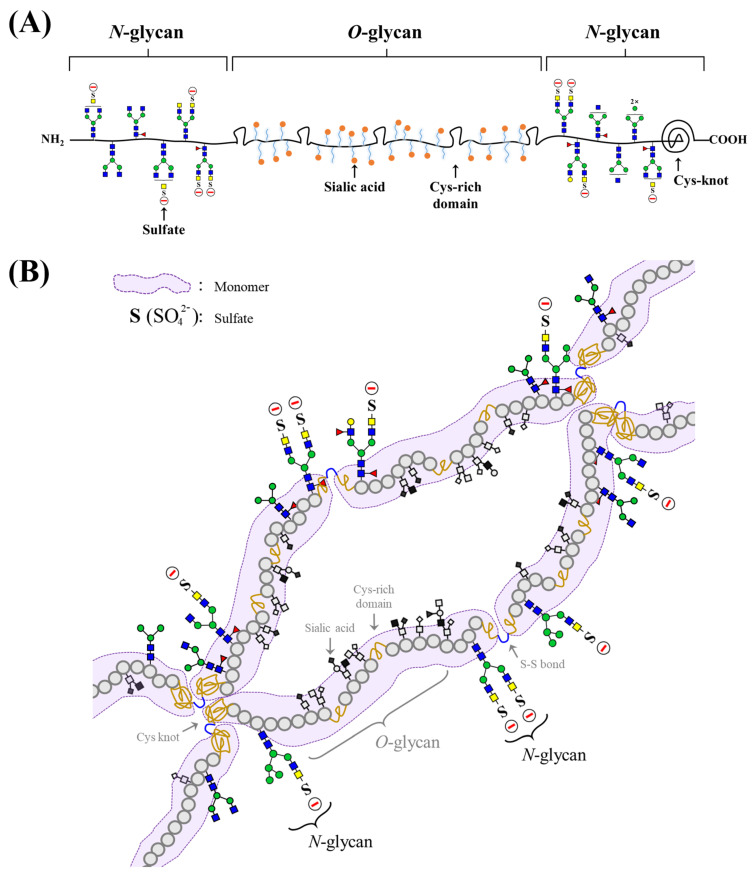
(**A**) Monomer and (**B**) polymer formation of BSM based on the present results (sulfated *N*-glycans) and previous reports (*O*-glycans, Cys-rich domain, Cys-knot, and S-S-bond) [12,14].

**Table 1 polymers-13-00103-t001:** Summary of structures, mass values, and relative quantities (%) of 32 ProA-labeled BSM *N*-glycans.

Group	Sub-Group(%) ^a^	Proposed Structure ^b^	Mass (*m/z*)	Relative Quantity (%) ^e^	Peak No ^f^
Calculated [M + H]^+^	Observed [M + H]^+ c^	Error ^d^ (ppm)
12sulfated *N*-glycans(27.9%)	5sulfated *N*-glycans(6.2%)	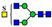	910.3554	910.3557 ^g^	0.3	1.9	7
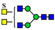	1011.8951	1011.8954 ^g^	0.3	1.6	10
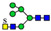	970.8685	970.8686 ^g^	0.1	1.2	23
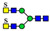	1051.8735	1051.8738 ^g^	0.3	0.8	9
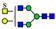	991.3818	991.3793 ^g^	−2.6	0.7	18
7sulfatedand core-fucosylated *N*-glycans(21.7%)	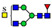	983.3844	983.3817 ^g^	−2.8	10.2	13
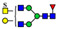	1064.4108	1064.4102 ^g^	−0.6	2.6	24
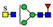	881.8447	881.8420 ^g^	−3.1	2.1	5
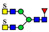	1124.9025	1124.9021 ^g^	−0.3	2.1	8
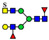	2273.8722	2273.8745	1.0	2.0	27
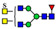	1084.9241	1084.9227 ^g^	−1.2	1.9	20
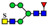	962.8711	962.8702 ^g^	−0.9	0.8	16
20non-sulfated *N*-glycans(72.1%)	11 core-fucosylated *N*-glycans(50.9%)	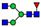	841.8663	841.8639 ^g^	−2.8	16.3	12
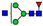	740.3266	740.3243 ^g^	−3.0	8.9	4
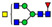	943.4060	943.4033 ^g^	−2.9	6.0	19
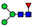	638.7869	638.7852 ^g^	−2.6	4.9	2
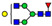	922.8927	922.8900 ^g^	−2.9	4.1	22
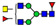	2031.8625	2031.8624	−0.1	3.0	30
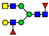	2193.9154	2193.9148	−0.3	2.1	32
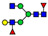	995.9216	995.9197 ^g^	−1.9	1.7	26
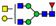	1024.4324	1024.4302 ^g^	−2.3	1.6	29
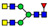	1044.9456	1044.9454 ^g^	−2.1	1.5	25
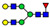	1003.9190	1003.9183 ^g^	−0.8	0.8	31
9 non-core-fucosylated *N*-glycans(21.2%)	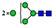	727.8108	727.8090 ^g^	−2.5	6.5	15
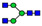	768.8373	768.8350 ^g^	−3.0	3.3	6
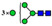	808.8372	808.8351 ^g^	−2.6	3.0	28
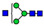	1333.5880	1333.5873	−0.5	2.2	3
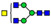	870.3770	870.3747 ^g^	−2.7	2.1	14
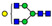	849.8637	849.8613 ^g^	−2.9	1.4	17
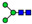	1130.5086	1130.5090	0.3	1.0	1
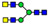	971.9167	971.9145 ^g^	−2.3	1.0	21
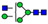	1495.6408	1495.6390	−1.3	0.7	11
		Sum		-		100.0	

^a^ Sum of relative quantities of each glycan in the group of total *N*-glycans. ^b^ Symbols: S, sulfate; ▲, Fuc; ■, GlcNAc; ■, GalNAc; ●, Gal; ●, Man. ^c^ Precursor ion mass obtained from full scan mass spectra. ^d^ Mass error was calculated as [(observed mass − calculated mass)/(calculated mass)] × 10^6^. ^e^ Relative quantity (%) was determined from EIC areas for all observed charge states of each glycan relative to total EIC area for all *N*-glycans. ^f^ The numbers correspond to the peak numbers in Figure 1. ^g^ Monoisotopic mass was detected as doubly-charged precursor ion [M + 2H]^2+^. Others were detected as singly-charged precursor ion [M + H]^+^.

## Data Availability

Data available in a publicly accessible repository.

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
