# Peer review of "N-Glycan Modifications with Negative Charge in a Natural Polymer Mucin from Bovine Submaxillary Glands, and Their Structural Role"

_polymers, 2020, doi:10.3390/polym13010103_

Round 1
Reviewer 1 Report
Authors present a study on the N-glycosylation in the bovine submaxillary mucin by using UPLC-MS and MS/MS of the procainamide labeled N-glycans released by PNGase F. This derivatization method proved to be an excellent choice, as sulfated glycans with a negative charge give a very poor response with other methods, as the authors themselves have seen.
The manuscript can be published after some modifications.
1) the definition used in table 1 is not good. Ok sulfated and not sulfated for the group but neutral N-glycans for the not core fucosylated N-glycans is not good. Better “not core fucosylated”.
2) Again in table 1, better “calculated” instead of “theoretical” in column 4.
3) In figure 1a there is an uninterpreted peak. If the value of m/z of this peak is, as it seems, 1495.6 it probably corresponds to a consecutive loss of sulphated GalNac and a GlcNac.
Author Response
Response to reviewer
Manuscript ID: polymers-1034054
Title: N-glycan Modifications with Negative Charge in a Natural Polymer Mucin from Bovine Submaxillary Glands, and Their Structural Role
[1] Reviewer #1
We appreciate the comments of Reviewer 1, and we fully agree with all of them. The revised version of the manuscript complies with each of this reviewer’s suggestions as follows:
Point 1: the definition used in table 1 is not good. Ok sulfated and not sulfated for the group but neutral N-glycans for the not core fucosylated N-glycans is not good. Better “not core fucosylated”.
Response 1: As suggested by the reviewer, “neutral” N-glycans has been changed to “non-core-fucosylated” N-glycans in Table 1, and two more sections containing the term have been revised as follows (additions in red bold and underlined):
(Revised) Table 1.
[Line 184]
|
Group |
Sub-group (%)a |
Proposed structureb |
Mass (m/z) |
Relative quantity (%)e |
Peak Nof |
||
|
Calculated [M+H]+ |
Observed [M+H]+c |
Errord (ppm) |
|||||
|
12 sulfated N-glycans (27.9%) |
5 sulfated N-glycans (6.2%) |
910.3554 |
910.3557g |
0.3 |
1.9 |
7 |
|
|
1011.8951 |
1011.8954g |
0.3 |
1.6 |
10 |
|||
|
970.8685 |
970.8686g |
0.1 |
1.2 |
23 |
|||
|
1051.8735 |
1051.8738g |
0.3 |
0.8 |
9 |
|||
|
991.3818 |
991.3793g |
-2.6 |
0.7 |
18 |
|||
|
7 sulfated and core-fucosylated N-glycans (21.7%) |
983.3844 |
983.3817g |
-2.8 |
10.2 |
13 |
||
|
1064.4108 |
1064.4102g |
-0.6 |
2.6 |
24 |
|||
|
881.8447 |
881.8420g |
-3.1 |
2.1 |
5 |
|||
|
1124.9025 |
1124.9021g |
-0.3 |
2.1 |
8 |
|||
|
2273.8722 |
2273.8745 |
1.0 |
2.0 |
27 |
|||
|
1084.9241 |
1084.9227g |
-1.2 |
1.9 |
20 |
|||
|
962.8711 |
962.8702g |
-0.9 |
0.8 |
16 |
|||
|
20 non-sulfated N-glycans (72.1%) |
11 core-fucosylated N-glycans (50.9%) |
841.8663 |
841.8639g |
-2.8 |
16.3 |
12 |
|
|
740.3266 |
740.3243g |
-3.0 |
8.9 |
4 |
|||
|
943.4060 |
943.4033g |
-2.9 |
6.0 |
19 |
|||
|
638.7869 |
638.7852g |
-2.6 |
4.9 |
2 |
|||
|
922.8927 |
922.8900g |
-2.9 |
4.1 |
22 |
|||
|
2031.8625 |
2031.8624 |
-0.1 |
3.0 |
30 |
|||
|
2193.9154 |
2193.9148 |
-0.3 |
2.1 |
32 |
|||
|
995.9216 |
995.9197g |
-1.9 |
1.7 |
26 |
|||
|
1024.4324 |
1024.4302g |
-2.3 |
1.6 |
29 |
|||
|
1044.9456 |
1044.9454g |
-2.1 |
1.5 |
25 |
|||
|
1003.9190 |
1003.9183g |
-0.8 |
0.8 |
31 |
|||
|
9 non-core-fucosylated N-glycans (21.2%) |
727.8108 |
727.8090g |
-2.5 |
6.5 |
15 |
||
|
768.8373 |
768.8350g |
-3.0 |
3.3 |
6 |
|||
|
808.8372 |
808.8351g |
-2.6 |
3.0 |
28 |
|||
|
1333.5880 |
1333.5873 |
-0.5 |
2.2 |
3 |
|||
|
870.3770 |
870.3747g |
-2.7 |
2.1 |
14 |
|||
|
849.8637 |
849.8613g |
-2.9 |
1.4 |
17 |
|||
|
1130.5086 |
1130.5090 |
0.3 |
1.0 |
1 |
|||
|
971.9167 |
971.9145g |
-2.3 |
1.0 |
21 |
|||
|
1495.6408 |
1495.6390 |
-1.3 |
0.7 |
11 |
|||
|
|
|
Sum |
|
- |
|
100.0 |
|
[Line 225]
The 32 N-glycans included 12 sulfated (27.9%, sum of relative quantities of each glycan) and 20 non-sulfated or non-core-fucosylated (72.1%) N-glycans. Their characterization depended on sulfation and/or core-fucosylation, such as 5 sulfated N-glycans (S1~2Fuc0HexNAc4~6Hex3~5 ; 6.2%), 7 sulfated and core-fucosylated N-glycans (S1~2Fuc1~2HexNAc4~6Hex3~4 ; 21.7%), 11 core-fucosylated N-glycans (S0Fuc1~2HexNAc2~6Hex3~5 ; 50.9%), and 9 non-sulfated and non-fucosylated N-glycans (S0Fuc0HexNAc2~6Hex3~6 ; 21.2%).
[Line 352]
Supplementary Materials: The following are available online at www.mdpi.com/xxx/s1. Figure S1: Full scan mass spectra of (A, B, C, D, and E) sulfated (peaks 7, 10, 23, 9, and 18), (F, G, H, I, J, K. and L) sulfated and core-fucosylated (peaks 13, 24, 5, 8, 27, 20, and 16), (M, N, O, P, Q, R, S, T, U, V, and W) core-fucosylated (peaks 12, 4, 19, 2, 22, 30, 32, 26, 29, 26, and 31), and (X, Y, Z, AA, AB, AC, AD, AE, and AF) non-core-fucosylated (peaks 15, 6, 28, 3, 14, 17, 1, 21, and 11) ProA-labeled N-glycans from BSM. Each peak number corresponds to a peak number in Table 1 and Figure 1. Precursor ions were detected as doubly-charged ion [M+2H]2+ and singly-charged ion [M+H]+. Figure S2: HCD-MS/MS spectra of (A, B, C, and D) sulfated (peaks 7, 10, 23, and 9), (E, F, G, and H) sulfated and core-fucosylated (peaks 24, 5, 8, and 27), (I and J) core-fucosylated (peaks 32 and 26), and (K and L) non-core-fucosylated (peaks 14 and 17) ProA-labeled N-glycans from BSM. Twenty N-glycans included 3 N-glycans (indicated in Figure 2) and 17 N-glycans (same glycan structures as in a previous report [19]) were not included. Each peak number corresponds to a peak number in Table 1 and Figure 1. Symbols: S, sulfate; ▲, Fuc; ■, GlcNAc; ■, GalNAc; ●, Gal; ●, Man.
Point 2: Again in table 1, better “calculated” instead of “theoretical” in column 4.
Response 2: As suggested by the reviewer, “theoretical” has been changed to “calculated” in Table 1 as well as three more sections containing the same word (additions in red, bold, and underlined):
(Revised) Table 1.
[Line 184]
|
Group |
Sub-group (%)a |
Proposed structureb |
Mass (m/z) |
Relative quantity (%)e |
Peak Nof |
||
|
Calculated [M+H]+ |
Observed [M+H]+c |
Errord (ppm) |
|||||
|
12 sulfated N-glycans (27.9%) |
5 sulfated N-glycans (6.2%) |
910.3554 |
910.3557g |
0.3 |
1.9 |
7 |
|
|
1011.8951 |
1011.8954g |
0.3 |
1.6 |
10 |
|||
|
970.8685 |
970.8686g |
0.1 |
1.2 |
23 |
|||
|
1051.8735 |
1051.8738g |
0.3 |
0.8 |
9 |
|||
|
991.3818 |
991.3793g |
-2.6 |
0.7 |
18 |
|||
|
7 sulfated and core-fucosylated N-glycans (21.7%) |
983.3844 |
983.3817g |
-2.8 |
10.2 |
13 |
||
|
1064.4108 |
1064.4102g |
-0.6 |
2.6 |
24 |
|||
|
881.8447 |
881.8420g |
-3.1 |
2.1 |
5 |
|||
|
1124.9025 |
1124.9021g |
-0.3 |
2.1 |
8 |
|||
|
2273.8722 |
2273.8745 |
1.0 |
2.0 |
27 |
|||
|
1084.9241 |
1084.9227g |
-1.2 |
1.9 |
20 |
|||
|
962.8711 |
962.8702g |
-0.9 |
0.8 |
16 |
|||
|
20 non-sulfated N-glycans (72.1%) |
11 core-fucosylated N-glycans (50.9%) |
841.8663 |
841.8639g |
-2.8 |
16.3 |
12 |
|
|
740.3266 |
740.3243g |
-3.0 |
8.9 |
4 |
|||
|
943.4060 |
943.4033g |
-2.9 |
6.0 |
19 |
|||
|
638.7869 |
638.7852g |
-2.6 |
4.9 |
2 |
|||
|
922.8927 |
922.8900g |
-2.9 |
4.1 |
22 |
|||
|
2031.8625 |
2031.8624 |
-0.1 |
3.0 |
30 |
|||
|
2193.9154 |
2193.9148 |
-0.3 |
2.1 |
32 |
|||
|
995.9216 |
995.9197g |
-1.9 |
1.7 |
26 |
|||
|
1024.4324 |
1024.4302g |
-2.3 |
1.6 |
29 |
|||
|
1044.9456 |
1044.9454g |
-2.1 |
1.5 |
25 |
|||
|
1003.9190 |
1003.9183g |
-0.8 |
0.8 |
31 |
|||
|
9 non-core-fucosylated N-glycans (21.2%) |
727.8108 |
727.8090g |
-2.5 |
6.5 |
15 |
||
|
768.8373 |
768.8350g |
-3.0 |
3.3 |
6 |
|||
|
808.8372 |
808.8351g |
-2.6 |
3.0 |
28 |
|||
|
1333.5880 |
1333.5873 |
-0.5 |
2.2 |
3 |
|||
|
870.3770 |
870.3747g |
-2.7 |
2.1 |
14 |
|||
|
849.8637 |
849.8613g |
-2.9 |
1.4 |
17 |
|||
|
1130.5086 |
1130.5090 |
0.3 |
1.0 |
1 |
|||
|
971.9167 |
971.9145g |
-2.3 |
1.0 |
21 |
|||
|
1495.6408 |
1495.6390 |
-1.3 |
0.7 |
11 |
|||
|
|
|
Sum |
|
- |
|
100.0 |
|
[Line 108]
LC-ESI-HCD-MS/MS was performed using an Ultimate 3000 LC system (Thermo Scientific, Waltham, MA, USA) coupled with a mass spectrometer (Q Exactive Hybrid Quadrupole-Orbitrap; Thermo Scientific). The same column and gradient were used for LC-ESI-HCD-MS/MS as for UPLC. The MS measurement was performed in positive ion mode over a range of m/z 500, and the HCD-MS/MS spectra were acquired over m/z 100 to observe N-glycan-generated fragment ions. The detailed mass spectrometer setting for MS analyses were as follows; full scan (m/z 500–3000) used resolution 70,000 with automatic gain control target of 1 × 106 ions and maximum injection time of 100 ms. Data-dependent MS/MS was performed on a “Top 5” data-dependent mode using the following parameters; resolution: 17,500, automatic gain control target: 2 × 105, maximum injection time: 50 ms. Source ionization parameters were as follows; spray voltage: 3.5 kV, capillary temperature: 320 °C, probe heater: 300 °C, and S-Lens level: 50. Data acquisition was controlled with an Xcalibur 2.1 (Thermo Fisher Scientific). Mass error was calculated as follows: [(observed mass – calculated mass) / (calculated mass)] × 106 [3]. All analytical experiments were repeated three times with identical LC chromatograms and MS ion chromatograms.
[Line 142]
N-glycan peaks were obtained from UPLC chromatograms (Figure 1A) and total ion chromatogram (TIC; Figure 1D) corresponding to 32 N-glycan structures (Figure S1), which were identified by EICs of two diagnostic oxonium ions [Sulfate(-SO4;S)1 N-acetylhexosamine(HexNAc)1 + H]+ (calculated m/z = 284.0435; Figure 1B) and [ProA-Fuc1HexNAc1 + H]+ (calculated m/z = 587.3287; Figure 1C).
[Line 190]
d Mass error was calculated as [(observed mass – calculated mass) / (calculated mass)] × 106.
Point 3: In figure 1a there is an uninterpreted peak. If the value of m/z of this peak is, as it seems, 1495.6 it probably corresponds to a consecutive loss of sulfated GalNac and a GlcNac
Response 3: As suggested by the reviewer, we have inserted the m/z value of the fragment ion peak in Figure 2A. It is identified as [ProA-HexNAc4Hex4 + H]+ (observed m/z = 1495.6399), and it is indicated by a red arrow.
(Revised) Figure 2.
[Line 172]
All of the modifications made in accordance with the reviewer’s suggestions are included in the fully revised version of the manuscript attached to this message. Thank you very much for considering our revised manuscript for publication in Polymers.
Yours sincerely,
Ha Hyung Kim, Professor, Ph.D.
Biotherapeutics and Glycomics Laboratory
College of Pharmacy, Chung-Ang University
84 Heukseok-ro, Dongjak-gu, Seoul 06944, South Korea
Tel.: 82-2-820-5612 fax: 82-2-823-5612 E-mail: hahyung@cau.ac.kr

Reviewer 2 Report
This manuscript describes a glycomic analysis of N-glycans, enzymatically released by PNG´ase F, from commercially available bovine submaxillary mucin (BSM-Type 1a), conjugated with procainamide and analyzed with UPLC and HPLC-ESI-MS/MS in the positive mode using HCD. All in all 32 N-linked glycans were identified and quantitated both in relative amounts and in absolute amounts. Twelve of these glycans were found sulfated, mainly monosulfated, and their core GlcNAc residue was found with and without fucose. No sialic acids, nor any phosphates were identified. From the results the authors discuss the potential biological role/s these N-glycans may play in forming the mucous network typically covering epithelial surfaces.
I think the aim of the study is very clear and interesting but the results provided makes me feel uncertain about the conclusions.
Major issues;
Already from the beginning of the review process I note that original MS data are missing and definitely not available for scrutinized evaluation. If the journal would like to follow the MIRAGE standard for presenting Glycomics MS data such data must be made available (Kolarich D, Rapp E, Struwe WB, Haslam SM, Zaia J, McBride R, Agravat S, Campbell MP, Kato M, Ranzinger R, Kettner C, York WS. The minimum information required for a glycomics experiment (MIRAGE) project: improving the standards for reporting mass-spectrometry-based glycoanalytic data. Mol Cell Proteomics. 2013 Apr;12(4):991-5. doi: 10.1074/mcp.O112.026492).
Secondly, I would say that it is generally accepted that for the analysis of sulfated glycoconjugates the MS analyses should be run in the negative mode and not in positive mode as done here. In positive mode sulfate groups are easily lost or migrate in the carbohydrate chains. If the ProA conjugates have another mode of ionization and stabilizes the sulfate modifications it should be clearly shown here - or at least discussed with reference to other publications. Preferentially, in this very publication a direct comparison of the two modes of MS-analysis should have been included.
Additionally, there is no description of how the original MS data were analyzed and which software/s were used and how their accuracies were validated. The interpretation of some of the MS data presented are questionable (see below).
As to the quantification it is not satisfactory described in the manuscript how the UPLC fluorescence data (not good enough resolution?) correlated to MS data. Additionally no standard curve is given for the MS-data.
Specific issues;
On lines 102 and 103 the m/z ranges must be printed wrong - since the spectra included give mass ranges up to m/z 1700.
The HCD level/s are not specified - which is critical for the fragmentation analysis
Fig.1 , There is no correlation described nor shown between the UPLC chromatogram and the TIC of the LC-MS analysis. That must be included.
Fig. 1, For the EIC of panel B - the major peak at around 22.5 min is not explained- why not? Should be.
Fig. 2 The peaks at m/z 366 and 1499-1500 are not annotated. The one at m/z 366 is found in all three panels and its interpretation as given in panel C is dubious and could very well or according to my experience preferentially be Gal-GlcNAc instead of what is explained as Man-GlcNAc from the core.
Fig. 2, The oxonium ion at m/z 284 is annotated as GalNAc-Sulfate but there is no trustworthy peak in panel A for this ion – compared to panel C, where it is supposed not to be, it is in the same range. How do you know the sulfate is on a GalNAc and not on any of the GlcNAc –especially since there are no other ions annotated with an additional GalNAc on one of the antennas?
Line 191-192 the authors state that there were unresolved peaks in the UPLC chromatogram which was not the case for the EIC – that must be shown not only as EICs but as TICS.
Fig.4 is a schematic illustration of mucin O-and N-glycosylation and its formation of networks. Nevertheless, there is no new experimental data supporting the distribution of N-glycans in this manuscript. All work presented is based on only glycomics data and nothing was done with glycoproteomics which, in its simplest case would have given information on which Asn:s that were indeed glycosylated and which were not. Since PNG´ase F was used that could be seen from the conversion of these Asn to Asp. To take the analysis a bit further it would have been very nice to analyze the distribution of different N-glycans at the N- and C-terminus of a specific mucin protein in the BSM through a LC-MS/MS based glycoproteomic analysis. Now this part of the manuscript remains mostly as speculations and should, including the whole paragraph 3.4 be transferred to the discussion paragraphs unless supported by additional solid experimental data.
Author Response
Response to reviewer
Manuscript ID: polymers-1034054
Title: N-glycan Modifications with Negative Charge in a Natural Polymer Mucin from Bovine Submaxillary Glands, and Their Structural Role
[2] Reviewer #2
We appreciate the comments of Reviewer 2, and we fully agree with all of them. See below the revised sections of the manuscript based on Reviewer 2’s suggestions:
Point 1: Already from the beginning of the review process I note that original MS data are missing and definitely not available for scrutinized evaluation. If the journal would like to follow the MIRAGE standard for presenting Glycomics MS data such data must be made available (Kolarich D, Rapp E, Struwe WB, Haslam SM, Zaia J, McBride R, Agravat S, Campbell MP, Kato M, Ranzinger R, Kettner C, York WS. The minimum information required for a glycomics experiment (MIRAGE) project: improving the standards for reporting mass-spectrometry-based glycoanalytic data. Mol Cell Proteomics. 2013 Apr;12(4):991-5. doi: 10.1074/mcp.O112.026492).
Response 1: In the present study, the glycan structures were identified in the following order using LC-MS: (1) the full scan mass spectrum for the precursor ion with its isotopes, (2) the MS/MS spectrum for fragmented ions generated from the precursor ion, and (3) a full scan mass spectrum for the multiple-charged precursor ion. At this time, all of the MS/MS spectra do not contain all fragment or precursor ions; i.e., the fragment or precursor ions may or may not be present in the MS/MS spectra [Ref 1]. Thus, the precursor ion should be identified by specific MS/MS fragment ions as well as the monoisotopic mass of the precursor ion in the full scan mass spectrum.
[Ref 1, not presented in the manuscript] De Vijlder, T.; Valkenborg, D.; Lemière, F.; Romijn, E.P.; Laukens, K.; Cuyckens, F. A tutorial in small molecule identification via electrospray ionization-mass spectrometry: The practical art of structural elucidation. Mass Spectrom. Rev. 2018, 37, 607–629.
As suggested by the reviewer, mass spectrometry is the most useful analytical approach in glycomics experiments. The MIRAGE standard provides the guidelines for instrumental hardware and data interpretation.
Therefore, we have inserted instrumental information on the hardware in the Materials and Methods 2.3 section. The software information for the full scan mass spectra (single-stage MS data) for glycans is presented in the Supplementary Materials (additions in red bold and underlined):
(Revised) 2. Materials and Methods
[Line 108]
LC-ESI-HCD-MS/MS was performed using an Ultimate 3000 LC system (Thermo Scientific, Waltham, MA, USA) coupled with a mass spectrometer (Q Exactive Hybrid Quadrupole-Orbitrap; Thermo Scientific). The same column and gradient were used for LC-ESI-HCD-MS/MS as for UPLC. The MS measurement was performed in positive ion mode over a range of m/z 500, and the HCD-MS/MS spectra were acquired over m/z 100 to observe N-glycan-generated fragment ions. The detailed mass spectrometer settings for MS analyses are discussed below. The full scan (m/z 500–3000) used 70,000 resolution with an automatic gain control target of 1 × 106 ions and a maximum injection time of 100 ms. Data-dependent MS/MS was performed in a “Top 5” data-dependent mode using the following parameters: 17,500 resolution, 2 × 105 automatic gain control target, and 50 ms maximum injection time. The source ionization was conducted with 3.5 kV spray voltage, 320 °C capillary temperature, 300 °C probe heater, and an S-Lens level of 50. The data acquisition was performed with an Xcalibur 2.1 (Thermo Fisher Scientific). Mass error was calculated as follows: [(observed mass – calculated mass) / (calculated mass)] × 106 [3]. All analytical experiments were repeated three times with identical LC chromatograms and MS ion chromatograms.
(Revised) Supplementary Materials.
[Line 352]
Supplementary Materials: The following data are available online at www.mdpi.com/xxx/s1. Figure S1: Full scan mass spectra of (A-E) sulfated (peaks 7, 10, 23, 9, and 18, respectively), (F- L) sulfated and core-fucosylated (peaks 13, 24, 5, 8, 27, 20, and 16, respectively), (M-W) core-fucosylated (peaks 12, 4, 19, 2, 22, 30, 32, 26, 29, 26, and 31, respectively), and (X-AF) non-core-fucosylated (peaks 15, 6, 28, 3, 14, 17, 1, 21, and 11, respectively) ProA-labeled N-glycans from BSM. Each peak number corresponds to a peak number in Table 1 and Figure 1. Precursor ions were detected as doubly-charged ions [M+2H]2+ and singly-charged ions [M+H]+.
Point 2: Secondly, I would say that it is generally accepted that for the analysis of sulfated glycoconjugates the MS analyses should be run in the negative mode and not in the positive mode as done here. In positive mode sulfate groups are easily lost or migrate in the carbohydrate chains. If the ProA conjugates have another mode of ionization and stabilize the sulfate modifications it should be clearly shown here - or at least discussed with reference to other publications. Preferentially, in this very publication a direct comparison of the two modes of MS-analysis should have been included.
Response 2: As suggested by the reviewer, analysis of the sulfated glycans in negative mode is generally preferred because it reduces the loss of native glycans with an anion terminal. However, the negative mode has disadvantages such as limited sensitivity due to high background and distortion from chloride isotopes [Ref 2]. The sulfated glycans are preserved during the analysis in ESI mode [Ref 3], and the ProA-labeling used in this study was developed to improve glycan ionization efficiency by adding a tertiary amine [Ref 4]. This makes it possible to recover suppressed sulfated glycans and contribute to structural determination through the formation of useful reporter ions in the cationized (metal) ion adducts.
[Ref 2, not presented in the manuscript] Harvey, D.J. Structural determination of N-linked glycans by matrix-assisted laser desorption/ionization and electrospray ionization mass spectrometry. Proteomics 2005, 5, 1774–1786.
[Ref 3, not presented in the manuscript] Lei, M.; Mechref, Y.; Novotny, M.V. Structural analysis of sulfated glycans by sequential double-permethylation using methyl iodide and deuteromethyl iodide. J. Am. Soc. Mass Spectrom. 2009, 20, 1660–1671.
[Ref 4, not presented in the manuscript] Zhou, S.; Veillon, L.; Dong, X.; Huang, Y.; Mechref, Y. Direct comparison of derivatization strategies for LC-MS/MS analysis of N-glycans. Analyst 2017, 142, 4446–4455.
Point 3: Additionally, there is no description of how the original MS data were analyzed and which software/s were used and how their accuracies were validated. The interpretation of some of the MS data presented are questionable (see below).
Response 3: We have inserted the analysis method of the original MS data into the Materials and Methods 2.3 section (additions in red, bold, and underlined):
(Revised) 2. Materials and Methods
[Line 108]
LC-ESI-HCD-MS/MS was performed using an Ultimate 3000 LC system (Thermo Scientific, Waltham, MA, USA) coupled with a mass spectrometer (Q Exactive Hybrid Quadrupole-Orbitrap; Thermo Scientific). The same column and gradient were used for LC-ESI-HCD-MS/MS as for UPLC. The MS measurement was performed in positive ion mode over a range of m/z 500, and the HCD-MS/MS spectra were acquired over m/z 100 to observe N-glycan-generated fragment ions. The detailed mass spectrometer settings for MS analyses are discussed below. The full scan (m/z 500–3000) used 70,000 resolution with an automatic gain control target of 1 × 106 ions and a maximum injection time of 100 ms. Data-dependent MS/MS was performed in a “Top 5” data-dependent mode using the following parameters: 17,500 resolution, 2 × 105 automatic gain control target, and 50 ms maximum injection time. The source ionization was conducted with 3.5 kV spray voltage, 320 °C capillary temperature, 300 °C probe heater, and an S-Lens level of 50. The data acquisition was performed with an Xcalibur 2.1 (Thermo Fisher Scientific). Mass error was calculated as follows: [(observed mass – calculated mass) / (calculated mass)] × 106 [3]. All analytical experiments were repeated three times with identical LC chromatograms and MS ion chromatograms.
In the present study, all N-glycans were obtained and interpreted by X-calibur software (Thermo Scientific). Although there are currently various software to interpret glycan MS data, it is still difficult to fully interpret glycans. Most of the existing software do not deal with monosaccharide definitions, and because of this limitation, they rely on manual data processing [Ref 5]. Therefore, to increase the reliability of the analysis, we analyzed and interpreted the MS data in the following order: (1) the nine precursor ion forms of glycan in the full scan mass spectrum resulting from various adduct ions, comprising three singly charged forms ([M + H]+, [M + Na]+, and [M + K]+) and six doubly charged forms ([M + 2H]2+, [M + H + Na] 2+, [M + H + K] 2+, [M + 2Na]2+, [M + Na + K]2+, and [M + 2K]2+); (2) the oxonium ions derived from glycans in the MS/MS spectrum; and (3) the mass value error was limited within 5 ppm, resulting in reliable data.
[Ref 5, not presented in the manuscript] Liu, G.; Cheng, K.; Lo, C.Y.; Li, J.; Qu, J.; Neelamegham, S. A comprehensive, open-source platform for mass spectrometry-based glycoproteomics data analysis. Mol. Cell Proteomics 2017, 16, 2032–2047.
Point 4: As to the quantification it is not satisfactory described in the manuscript how the UPLC fluorescence data (not good enough resolution?) correlated to MS data. Additionally no standard curve is given for the MS-data
Response 4: As suggested, we have inserted the correlation of UPLC fluorescence data and MS data into the Result 3.1. section and Result 3.3 section (additions in red, bold, and underlined):
(Revised) 3. Result
[Line 142]
N-glycan peaks were obtained from UPLC chromatograms (Figure 1A) and the total ion chromatogram (TIC; Figure 1D) corresponding to 32 N-glycan structures (Figure S1), which were identified by the EICs of two diagnostic oxonium ions: [Sulfate(-SO4;S)1 N-acetylhexosamine(HexNAc)1 + H]+ (calculated m/z = 284.0435; Figure 1B) and [ProA-Fuc1HexNAc1 + H]+ (calculated m/z = 587.3287; Figure 1C).
(Revised) Figure 1.
[Line 154]
Figure 1. (A) UPLC chromatogram, (B) EIC of diagnostic oxonium ion for sulfation ([S1HexNAc1 + H]+, m/z 284.0435), (C) EIC of diagnostic oxonium ion for core-fucosylation ([ProA-Fuc1HexNAc1 + H]+ , m/z 587.3287), and (D) TIC of ProA-labeled N-glycans from BSM. The peak numbers correspond to N-glycans identified using LC-ESI-HCDMS/MS.
(already included in the manuscript) 3. Results
[Line 220]
The area of each peak of the UPLC chromatogram usually indicates the glycan quantity. However, in this study, the quantity (%) of each N-glycan relative to the total N-glycan amount (100%) was characterized according to the EIC rather than the UPLC chromatogram, as there were some overlapping peaks in the UPLC chromatogram, unlike in the EIC extracted as a single peak.
In addition, we have calculated glycan abundance using both UPLC-FLR and LC-ESI-HCD-MS/MS for the following reasons. Since multiple glycans can exist on one peak in the UPLC-FLR chromatogram, it could not distinguish the co-eluted glycans. Thus, analysis for both qualification and quantitation is not possible. Meanwhile, in the case of LC-ESI-HCD-MS/MS, despite the difference in the degree of ionization for each glycan, it is suitable for qualification because all glycans can be extracted as a single peak, and their areas also are confirmed by X-calibur software (Thermo Scientific). Therefore, the relative quantity and absolute quantity are calculated based on MS/MS and FLR, respectively.
The relative quantity and absolute quantity used in this study are based on fluorescence intensity and MS/MS ionization. We obtained the calibration curve for ProA concentration (0.1–12.5 pmol) using FLR, which was generated by the equation ‘y = 39.547 x+3.9533’ with r2 = 0.99 considered reliable data (Fig. 1).
Fig. 1 (not presented in the manuscript). Calibration curve for ProA concentration (0.1–12.5 pmol) using FLR
The absolute quantities of N-glycans were calculated as a ratio of the relative quantities in light of the two distinctively separated glycan peaks (sulfated or not), which are used as standard groups (peak 13 [sulfated N-glycan] and peak 2 [non-sulfated N-glycan]). This information is already mentioned in the Discussion as follows.
(already included in the manuscript) 4. Discussion
[Line 313]
The stoichiometric ratio of ProA to glycan is reportedly 1:1 [38]. Therefore, glycan concentrations were estimated based on the fluorescence intensities of the negatively charged (peak 13 in Figure 1) and neutral (peak 2 in Figure 1) glycan peaks using a linear calibration curve (r2 = 0.99) generated from the ProA concentrations (0.1–12.5 pmol; data not shown). The concentration range of each unreported N-glycan was 0.2–0.8 pmol. The total concentration, which was the sum of 12 negatively charged (8.2 pmol) and 20 neutral (21.0 pmol) N-glycans, was 29.2 pmol in 20 µg of BSM.
Point 5: On lines 102 and 103 the m/z ranges must be printed wrong - since the spectra included give mass ranges up to m/z 1700
Response 5: We agree with the reviewer regarding the m/z value used as the minimum value in the manuscript, which is difficult for readers to understand. Therefore, we have inserted the detailed information on the m/z value range into the Materials and Methods 2.3. section (additions in red, bold, and underlined):
(Revised) 2. Materials and Methods
[Line 108]
LC-ESI-HCD-MS/MS was performed using an Ultimate 3000 LC system (Thermo Scientific, Waltham, MA, USA) coupled with a mass spectrometer (Q Exactive Hybrid Quadrupole-Orbitrap; Thermo Scientific). The same column and gradient were used for LC-ESI-HCD-MS/MS as for UPLC. The MS measurement was performed in positive ion mode over a range of m/z 500, and the HCD-MS/MS spectra were acquired over m/z 100 to observe N-glycan-generated fragment ions. The detailed mass spectrometer settings for MS analyses are discussed below. The full scan (m/z 500–3000) used 70,000 resolution with an automatic gain control target of 1 × 106 ions and a maximum injection time of 100 ms. Data-dependent MS/MS was performed in a “Top 5” data-dependent mode using the following parameters: 17,500 resolution, 2 × 105 automatic gain control target, and 50 ms maximum injection time. The source ionization was conducted with 3.5 kV spray voltage, 320 °C capillary temperature, 300 °C probe heater, and an S-Lens level of 50. The data acquisition was performed with an Xcalibur 2.1 (Thermo Fisher Scientific). Mass error was calculated as follows: [(observed mass – calculated mass) / (calculated mass)] × 106 [3]. All analytical experiments were repeated three times with identical LC chromatograms and MS ion chromatograms.
Point 6: The HCD level/s are not specified - which is critical for the fragmentation analysis.
Response 6: As suggested, we have inserted the detailed HCD level and other parameters into the Materials and Methods 2.3. section as follows (additions in red, bold, and underlined):
(Revised) 2. Materials and Methods
[Line 108]
LC-ESI-HCD-MS/MS was performed using an Ultimate 3000 LC system (Thermo Scientific, Waltham, MA, USA) coupled with a mass spectrometer (Q Exactive Hybrid Quadrupole-Orbitrap; Thermo Scientific). The same column and gradient were used for LC-ESI-HCD-MS/MS as for UPLC. The MS measurement was performed in positive ion mode over a range of m/z 500, and the HCD-MS/MS spectra were acquired over m/z 100 to observe N-glycan-generated fragment ions. The detailed mass spectrometer settings for MS analyses are discussed below. The full scan (m/z 500–3000) used 70,000 resolution with an automatic gain control target of 1 × 106 ions and a maximum injection time of 100 ms. Data-dependent MS/MS was performed in a “Top 5” data-dependent mode using the following parameters: 17,500 resolution, 2 × 105 automatic gain control target, and 50 ms maximum injection time. The source ionization was conducted with 3.5 kV spray voltage, 320 °C capillary temperature, 300 °C probe heater, and an S-Lens level of 50. The data acquisition was performed with an Xcalibur 2.1 (Thermo Fisher Scientific). Mass error was calculated as follows: [(observed mass – calculated mass) / (calculated mass)] × 106 [3]. All analytical experiments were repeated three times with identical LC chromatograms and MS ion chromatograms.
Point 7: Fig.1, There is no correlation described nor shown between the UPLC chromatogram and the TIC of the LC-MS analysis. That must be included.
Response 7: We used extracted ion chromatogram (EIC) of sulfated or core-fucosylated glycans in this study, which were extracted from their total ion chromatogram (TIC). It is difficult to distinguish specific peaks for glycans in TIC, because it displays all ions including solvents in one chromatogram. However, the peaks of EICs could be used to identify the glycan. Thus, the EICs of characteristic oxonium ions derived from glycan ([S1HexNAc1 + H]+ ion, calculated m/z = 284.0435 and [ProA-Fuc1HexNAc1 + H]+ , calculated m/z = 587.3287) were used with the UPLC chromatogram.
In addition, the N-glycan signal is indicated as fluorescence intensity in the UPLC and as abundance for degree of ionization in the MS, and glycans detected in the MS were eluted under the same gradient condition in LC. The ionization degree of the fluorescent derivatized part of glycan also has very high sensitivity in MS [Ref 6] and generates an improved MS signal [Ref 7].
[Ref 6, not presented in the manuscript] Manz, C.; Grabarics, M.; Hoberg, F.; Pugini, M.; Stuckmann, A.; Struwe, W.B.; Pagel, K. Separation of isomeric glycans by ion mobility spectrometry - the impact of fluorescent labelling. Analyst. 2019, 144, 5292–5298.
[Ref 7, not presented in the manuscript]
Nwosu, C.; Yau, H.K.; Becht, S. Assignment of core versus antenna fucosylation types in protein N‑glycosylation via procainamide labeling and tandem mass spectrometry. Anal. Chem. 2015, 87, 5905–5913.
As suggested by the reviewer, we have inserted the correlation between UPLC and the TIC of the LC-MS in the Results 3.1 section and Figure 1 (additions in red, bold, and underlined):
(Revised) 3. Results
[Line 142]
N-glycan peaks were obtained from UPLC chromatograms (Figure 1A) and the total ion chromatogram (TIC; Figure 1D) corresponding to 32 N-glycan structures (Figure S1), which were identified by the EICs of two diagnostic oxonium ions: [Sulfate(-SO4;S)1 N-acetylhexosamine(HexNAc)1 + H]+ (calculated m/z = 284.0435; Figure 1B) and [ProA-Fuc1HexNAc1 + H]+ (calculated m/z = 587.3287; Figure 1C).
(Revised) Figure 1.
[Line 154]
Figure 1. (A) UPLC chromatogram, (B) EIC of diagnostic oxonium ion for sulfation ([S1HexNAc1 + H]+, m/z 284.0435), (C) EIC of diagnostic oxonium ion for core-fucosylation ([ProA-Fuc1HexNAc1 + H]+ , m/z 587.3287), and (D) TIC of ProA-labeled N-glycans from BSM. The peak numbers correspond to N-glycans identified using LC-ESI-HCDMS/MS.
Point 8: Fig. 1, For the EIC of panel B - the major peak at around 22.5 min is not explained- why not? Should be.
Response 8: The EIC of the [S1HexNAc1 + H]+ ion showed the most abundant peak at 22.6 min, which is the glycan composed of S1Fuc1HexNAc3Hex3. However, since the ionization efficiency of sulfated ion was low in positive ion mode and the EIC itself shows a relative value rather than an absolute value, if the specific peak is slightly higher than others, it could appear as the abundant peak. Therefore, the peak at 22.6 min, the glycan (S1Fuc1HexNAc3Hex3), was excluded in the results because it has very low quantity in the total glycan pool.
Point 9: Fig. 2 The peaks at m/z 366 and 1499-1500 are not annotated. The one at m/z 366 is found in all three panels and its interpretation as given in panel C is dubious and could very well or according to my experience preferentially be Gal-GlcNAc instead of what is explained as Man-GlcNAc from the core.
Response 9: As suggested by the reviewer, the interpretation of Man-GlcNAc at m/z 366 could be ambiguous. Therefore, we have changed the galactose and/or mannose to hexose (as indicated by the white circle) in Figure 2 according to the previous report [Ref 8].
(Revised) Figure 2.
[Line 172]
[Ref 8, not presented in the manuscript] Eshghi, S.T.; Yang, W.; Hu, Y.; Shah, P.; Sun, S.; Li, X.; Zhang, H. Classification of tandem mass spectra for identification of N- and O-linked glycopeptides. Sci. Rep. 2016, 6, 37189.
Point 10: Fig. 2, The oxonium ion at m/z 284 is annotated as GalNAc-Sulfate but there is no trustworthy peak in panel A for this ion – compared to panel C, where it is supposed not to be, it is in the same range. How do you know the sulfate is on a GalNAc and not on any of the GlcNAc –especially since there are no other ions annotated with an additional GalNAc on one of the antennas?
Response 10: As mentioned by the reviewer, the oxonium ion [HexNAc-sulfate] at m/z 284.0435 in Figure 2A was not shown in the overall MS/MS spectrum because of the low intensity. Thus, we added the oxonium ion peak (calculated m/z = 484.1228) corresponding to [HexNAc2-sulfate] in the MS/MS spectrum. Additionally, it is difficult to distinguish between the GAlNAc-Sulfate and GlcNAc-Sulfate using MS, but it has been reported as the terminal GalNAc-Sulfate in bovine glycoprotein [Ref 9] and submaxillary gland glycoprotein [Ref 10], rather than the GlcNAc-Sulfate.
We have inserted the expanded version of the MS/MS spectrum (Fig. 2) and MS/MS spectra including an additional oxonium ion [S1HexNAc2 + H]+ at m/z 487.1234 and 487.1241 in Figure 2A and 2B as follows:
Fig. 2 (not presented in the manuscript). The expanded HCD-MS/MS spectra of (A) sulfated (peak 18) and (B) sulfated and core-fucosylated (peak 16) N-glycans with ProA-labeling.
[Ref 9, not presented in the manuscript] Yu, S.Y.; Chang, L.Y.; Cheng, C.W.; Chou, C.C.; Fukuda, M.N.; Khoo, K.H.; Priming mass spectrometry-based sulfoglycomic mapping for identification of terminal sulfated lacdiNAc glycotope. Glycoconj. J. 2012, 30, 183–194.
[Ref 10, not presented in the manuscript] Hooper, L.V.; Beranek, M.C.; Manzella, S.M.; Baenziger, J.U. Differential expression of GalNAc-4-sulfotransferase and GalNAc-transferase Results in distinct glycoforms of carbonic anhydrase VI in parotid and submaxillary glands. J. Biol. Chem. 1995, 270, 5985–5993.
(Revised) Figure 2.
[Line 172]
Point 11: Line 191-192 the authors state that there were unresolved peaks in the UPLC chromatogram which was not the case for the EIC – that must be shown not only as EICs but as TICS.
Response 11: As mentioned by the reviewer, there were unresolved peaks in the UPLC chromatogram that did not occur in the EIC. However, the UPLC chromatograms (Figure 1A), including 12 peaks from N-glycans with sulfation-containing diagnostic oxonium ions [Sulfate(-SO 4;S)1 N-acetylhexosamine(HexNAc)1 + H]+ (Figure 1B), and 18 peaks from N-glycans with fucosylation-containing diagnostic oxonium ions [ProA-Fuc1HexNAc1 + H]+ (Figure 1C), which have already mentioned in the manuscript.
Additionally, this study used the EIC of sulfated or core-fucosylated glycans, which were extracted from their TIC, respectively. It is difficult to distinguish specific glycan peaks from the TIC, because it displays all ions including solvents in one chromatogram. However, the peaks of the EICs could be used to identify the glycan. Thus, the EICs of characteristic oxonium ions derived from glycan ([S1HexNAc1 + H]+ ion, calculated m/z = 284.0435 and [ProA-Fuc1HexNAc1 + H]+ , calculated m/z = 587.3287) were used with UPLC chromatogram.
(Revised) Figure 1.
[Line 154]
(Revised) 3. Result
[Line 142]
N-glycan peaks were obtained from UPLC chromatograms (Figure 1A) and the total ion chromatogram (TIC; Figure 1D) corresponding to 32 N-glycan structures, which were identified by the EICs of two diagnostic oxonium ions: [Sulfate(-SO4;S)1 N-acetylhexosamine(HexNAc)1 + H]+ (calculated m/z = 284.0435; Figure 1B) and [ProA-Fuc1HexNAc1 + H]+ (calculated m/z = 587.3287; Figure 1C).
Point 12: Fig.4 is a schematic illustration of mucin O-and N-glycosylation and its formation of networks. Nevertheless, there is no new experimental data supporting the distribution of N-glycans in this manuscript. All work presented is based on only glycomics data and nothing was done with glycoproteomics which, in its simplest case would have given information on which Asn:s that were indeed glycosylated and which were not. Since PNG´ase F was used that could be seen from the conversion of these Asn to Asp. To take the analysis a bit further it would have been very nice to analyze the distribution of different N-glycans at the N- and C-terminus of a specific mucin protein in the BSM through a LC-MS/MS based glycoproteomic analysis. Now this part of the manuscript remains mostly as speculations and should, including the whole paragraph 3.4 be transferred to the discussion paragraphs unless supported by additional solid experimental data.
Response 12: As suggested by the reviewer, the paragraph for Figure 4 has been moved to the Discussion (Line 281):
(Revised) 4. Discussion
[Line 280]
Schematic illustrations of the mucin structures based on the present results and previous reports [12,14] are shown in Figure 4. The N-glycans, including the negatively charged sulfated N-glycans identified in this study, are located in the hydrophobic terminal regions (Figure 4A) [12,19].
The negatively charged glycans or positively charged amino acids (containing Lys, Arg, and His) in mucin might enhance its adsorption onto positive or negative surfaces in an aqueous environment, respectively. Sulfated N-glycans might contribute to the adsorption (or desorption) onto positive (or negative) surfaces due to attractive (or repulsive) forces, as well as hydration with water-holding and protection of degradation from foreign bacterial sialidases in mucin [12].
The C-terminal Cys-rich part (including the Cys knot) of mucin forms disulfide-bonded mucin dimers and multimers [32]. The present results suggest that the physicochemical characterization (e.g., pH sensitivity and stability) of BSM is derived from the sulfate (pKa = 1) groups in N-glycans as well as the reported sialic acid (pKa = 2.6) with some sulfate groups in O-glycans and charged amino acids [33]. Finally, sulfated N-glycans may structurally affect mucin mobility due to inter- and intramolecular electrostatic interactions, and associative polymer networks through self-association could form and stabilize together with hydrogen bonding (Figure 4B).
As mentioned by the reviewer, further research is needed to identify the glycosylation sites and the distribution of N-glycans in the N- and C-terminal of BSM based on the LC-MS/MS to conduct the glycoproteomic analysis.
All modifications made in accordance with the reviewer’s suggestions are included in the fully revised version of the manuscript attached to this message. Thank you very much for considering our revised manuscript for publication in Polymers.
Yours sincerely,
Ha Hyung Kim, Professor, Ph.D.
Biotherapeutics and Glycomics Laboratory
College of Pharmacy, Chung-Ang University
84 Heukseok-ro, Dongjak-gu, Seoul 06944, South Korea
Tel.: 82-2-820-5612 fax: 82-2-823-5612 E-mail: hahyung@cau.ac.kr
